# Catalytic enantioselective reductive domino alkyl arylation of acrylates via nickel/photoredox catalysis

Pengcheng Qian[1,5], Haixing Guan[1,2,5], Yan-En Wang[3], Qianqian Lu[1], Fan Zhang[1], Dan Xiong[1], Patrick J. Walsh ✉[4] & Jianyou Mao ✉[1]

Nonsteroidal anti-inflammatory drug derivatives (NSAIDs) are an important class of medications. Here we show a visible-light-promoted photoredox/nickel catalyzed approach to construct enantioenriched NSAIDs via a three-component alkyl arylation of acrylates. This reductive cross-electrophile coupling avoids preformed organometallic reagents and replaces stoichiometric metal reductants by an organic reductant (Hantzsch ester). A broad range of functional groups are well-tolerated under mild conditions with high enantioselectivities (up to 93% ee) and good yields (up to 90%). A study of the reaction mechanism, as well as literature precedence, enabled a working reaction mechanism to be presented. Key steps include a reduction of the alkyl bromide to the radical, Giese addition of the alkyl radical to the acrylate and capture of the α-carbonyl radical by the enantioenriched nickel catalyst. Reductive elimination from the proposed Ni(III) intermediate generates the product and forms Ni(I).

[1] Technical Institute of Fluorochemistry (TIF), Institute of Advanced Synthesis, School of Chemistry and Molecular Engineering, Nanjing Tech University, Nanjing, PR China. [2] Experimental Center, Key Laboratory of Traditional Chinese Medicine Classical Theory, Ministry of Education, Shandong University of Traditional Chinese Medicine, Jinan, PR China. [3] College of Science, Hebei Agricultural University, Baoding, PR China. [4] Roy and Diana Vagelos Laboratories, Department of Chemistry, University of Pennsylvania, Philadelphia, PA, USA. [5] These authors contributed equally: Pengcheng Qian, Haixing Guan. ✉email: pwalsh@sas.upenn.edu; ias_jymao@njtech.edu.cn

Enantioenriched α-aryl propionic acids are an important class of nonsteroidal anti-inflammatory medications (NSAIDs)[1,2] and are also key building blocks for further elaboration. As a result, significant effort has been devoted to the asymmetric synthesis of α-aryl carboxylic acid derivatives. Among enantioselective methods to prepare these structural motifs, asymmetric hydrogenation[3,4] and hydrocarboxylation[5] stand out. A more efficient approach to access such tertiary stereocenters, however, would be the transition metal catalyzed asymmetric α-arylation of esters.

The asymmetric cross-coupling between α-halo esters and organometallic reagents (e.g., Grignard, organozinc, organoboron, and organosilicon) (Fig. 1a) was developed and driven mainly by G.C. Fu and coworkers[6–9]. A complementary approach was reported by Zhou[10] and Gaunt[11] involving asymmetric arylation of enol silane derivatives with aryl sulfonates or iodonium salts under enantioselective transition metal catalysis (Fig. 1b). Motivated by a desire to broaden the scope of coupling partners while also avoiding moisture- and air-sensitive organometallic reagents, Reisman[12–16], Doyle[17], Weix[18], and their groups developed nickel catalyzed asymmetric reductive cross-electrophile coupling reactions using stoichiometric metal reductants (Zn or Mn). Inspired by their elegant studies, we disclosed an example of highly enantioselective nickel-photoredox catalyzed reductive cross-coupling of racemic α-chloro esters with

aryl iodides (Fig. 1c) to construct enantioenriched NSAID derivatives[19]. Building on this work, we envisioned intercepting intermediates along the enantioselective cross-electrophile coupling reaction pathway with an olefin insertion step. Such a strategy could potentially lead to valuable enantioenriched olefin difunctionalization products.

Recently, nickel-catalyzed reductive alkene difunctionalization reactions have gained notoriety as one of the most efficient strategies to install vicinal bonds in a single operation. Relevant examples were reported by Nevado[20,21], Chu[22,23], Martin[24], Yuan[25] and Molander[26,27], among others. Impressive enantioselective three-component reactions were developed by Diao[28], Chu[29,30], Nevado[31] as well as others[32–38]. In the final stages of preparing this manuscript, a complementary enantioselective three-component carboarylation of alkenes with alkyltrifluoroborates and aryl bromides was disclosed by Gutierrez and Chu[29] (Fig. 1d). The redox-neutrality of this method is counterbalanced by the use of alkyltrifluoroborates, which must be prepared separately.

In this work, we present an example of enantioselective domino alky arylation of acrylates with alkyl- and aryl- bromides via cooperative nickel and photoredox catalysis (Fig. 1e). The organic photoredox 1,2,3,5-tetrakis(carbazol-9-yl)-4,6-dicyanobenzene (4CzIPN) acts as "pseudoreductant" with Hantzsch ester (HEH) as the terminal

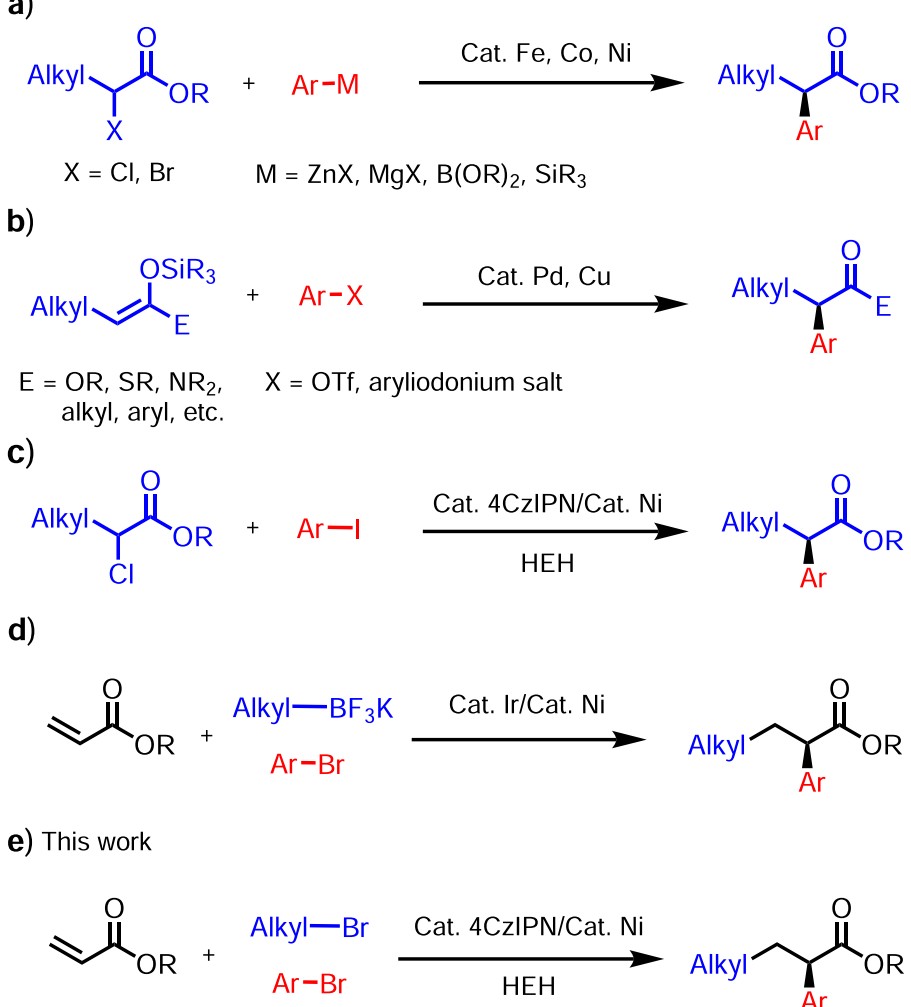

**Fig. 1 Transition-metal-catalyzed asymmetric coupling reactions to synthesize NSAID derivatives. a** The classical asymmetric cross-coupling with organometallic reagents. **b** Transition metal catalyzed asymmetric arylation of enol silane derivatives. **c** Nickel-photoredox co-catalyzed asymmetric reductive arylation of racemic α-chloro esters. **d** A complementary photoredox/nickel catalyzed enantioselective carboarylation of alkenes. **e** This work: Nickel/photoredox catalyzed reductive asymmetric alky arylation of acrylates.

**Table 1 Optimization of reaction conditions [a].**

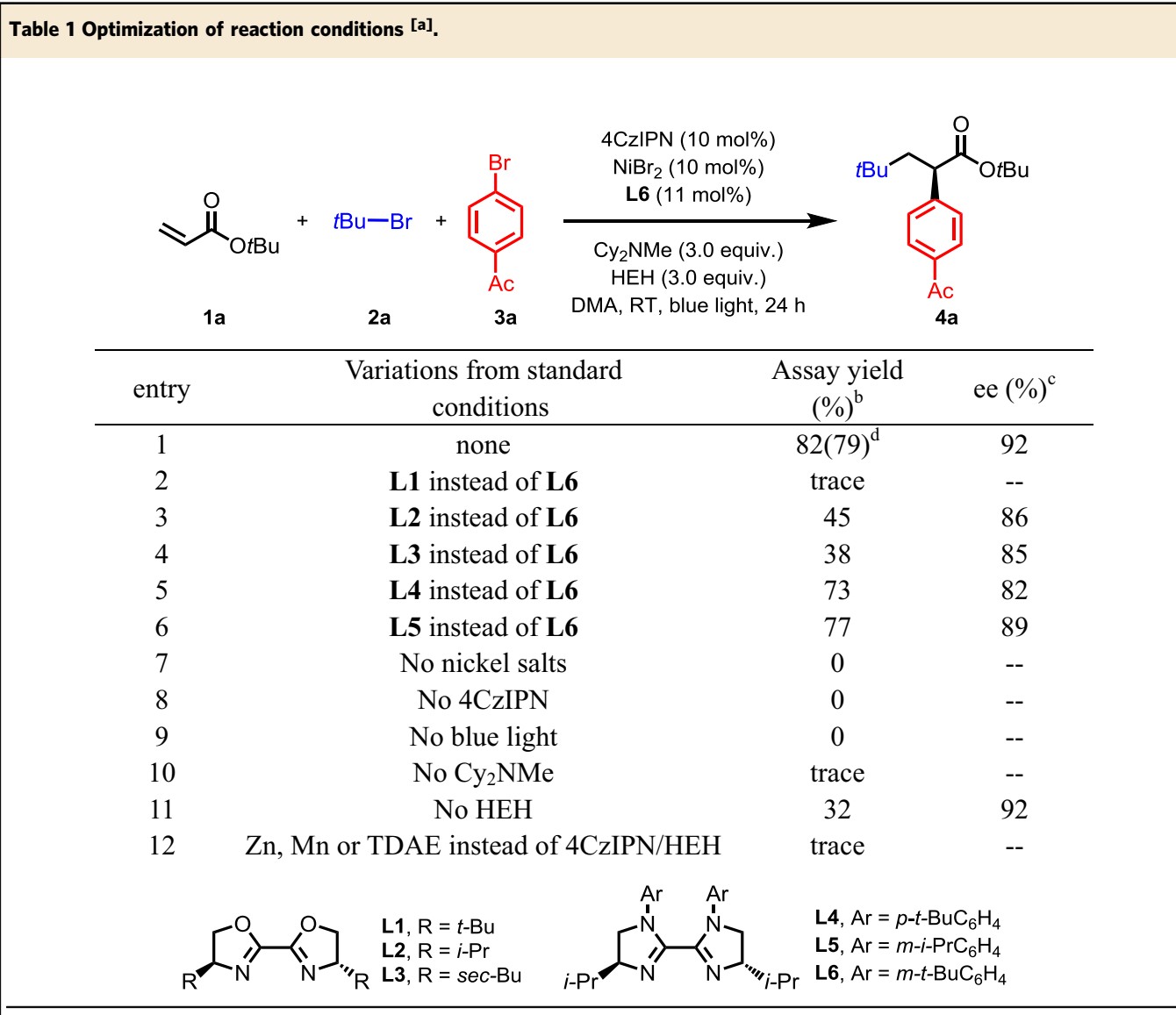

| entry | Variations from standard conditions | Assay yield (%)[b] | ee (%)[c] |
|---|---|---|---|
| 1 | none | 82(79)[d] | 92 |
| 2 | **L1** instead of **L6** | trace | -- |
| 3 | **L2** instead of **L6** | 45 | 86 |
| 4 | **L3** instead of **L6** | 38 | 85 |
| 5 | **L4** instead of **L6** | 73 | 82 |
| 6 | **L5** instead of **L6** | 77 | 89 |
| 7 | No nickel salts | 0 | -- |
| 8 | No 4CzIPN | 0 | -- |
| 9 | No blue light | 0 | -- |
| 10 | No Cy₂NMe | trace | -- |
| 11 | No HEH | 32 | 92 |
| 12 | Zn, Mn or TDAE instead of 4CzIPN/HEH | trace | -- |

**L1**, R = *t*-Bu
**L2**, R = *i*-Pr
**L3**, R = *sec*-Bu

**L4**, Ar = *p*-*t*-BuC₆H₄
**L5**, Ar = *m*-*i*-PrC₆H₄
**L6**, Ar = *m*-*t*-BuC₆H₄

[a] Reactions conducted under Ar on 0.1 mmol scale. [b] Determined by GC using tetradecane as an internal standard. [c] Determined by chiral HPLC on a CHIRALPAK IA column. [d] Isolated yield.

electron donor. The products of this cascade process are NSAID derivatives of potential use in medicinal chemistry.

## Results and discussion

**Reaction development and optimization.** We began our investigation into the three-component olefin difunctionalization reaction by choosing *tert*-butyl acrylate (**1a**), *tert*-butyl bromide (**2a**) and 1-(4-bromophenyl)ethan-1-one (**3a**) as model substrates (Table 1). After a systematic study of reaction conditions (see Table S1 in the Supporting Information for details), we were please to find that employing 4CzIPN (10 mol%), NiBr₂ (10 mol%), **L6** (11 mol%), Cy₂NMe (3.0 equiv.) and HEH (3.0 equiv.) in *N,N*-dimethylacetamide (DMA, 0.33 M) under blue LED irradiation at room temperature for 24 h furnished the α-aryl ester (**4a**) in 82% assay yield (AY, determined by GC integration of the unpurified reaction mixture against an internal standard). Gratifyingly, the ee value of **4a** was 92%. The (*R*) configuration of **4a** was confirmed by comparison of its optical rotation with the literature value[30] (see the Supporting Information for details). Based on our previous work[19], enantioenriched bioxazoline (BiOX) frameworks were considered promising ligands for the three component coupling, giving up to 86% ee. (Table 1. entry 2–3). Further optimization, however, indicated the

more electron-donating[39] chiral biimidazoline (BiIM) ligands were better both in yield and enantioselectivity. Control experiments indicated that nickel salts, 4CzIPN, Cy₂NMe and blue LEDs are all crucial for the success of this transformation (Table 1. entry 7–10). Additionally, HEH played a key role in the reaction yields. Only 32% AY of **4a** was detected in the absence of HEH (Table 1. entry 11), although, the enantioselectivity remained 92%. In this case, it is possible that the Cy₂NMe[40–42] plays the role of reducing agent, albeit with much reduced efficiency. When other reductants, such as Mn, Zn and tetrakis(dimethylamino)ethylene (TDAE) were employed, however, only trace products were detected. The diminished reactivity with Mn, Zn and TDAE highlight the advantage of HEH under blue light in this process.

**Substrate scope.** With the optimized reaction conditions in hand, we next focused on the scope of the aryl bromide coupling partners using *tert*-butyl acrylate (**1a**) and 4-*tert*-butyl bromide **2a** (Table 2). We were pleased to find that aryl bromides with functional groups in the 4-position, including electron-withdrawing groups (Ac, COOEt, CN, CHO, CF₃, OCF₃), halogens (F, Cl), neutral groups (H, Ph), and electron-donating groups (*t*Bu, OMe, SMe, NMe₂) all exhibited high to excellent *ee* values (83–93%) and moderate to good yields

**Table 2 Scope of aryl and (hetero)aryl bromides.[a].**

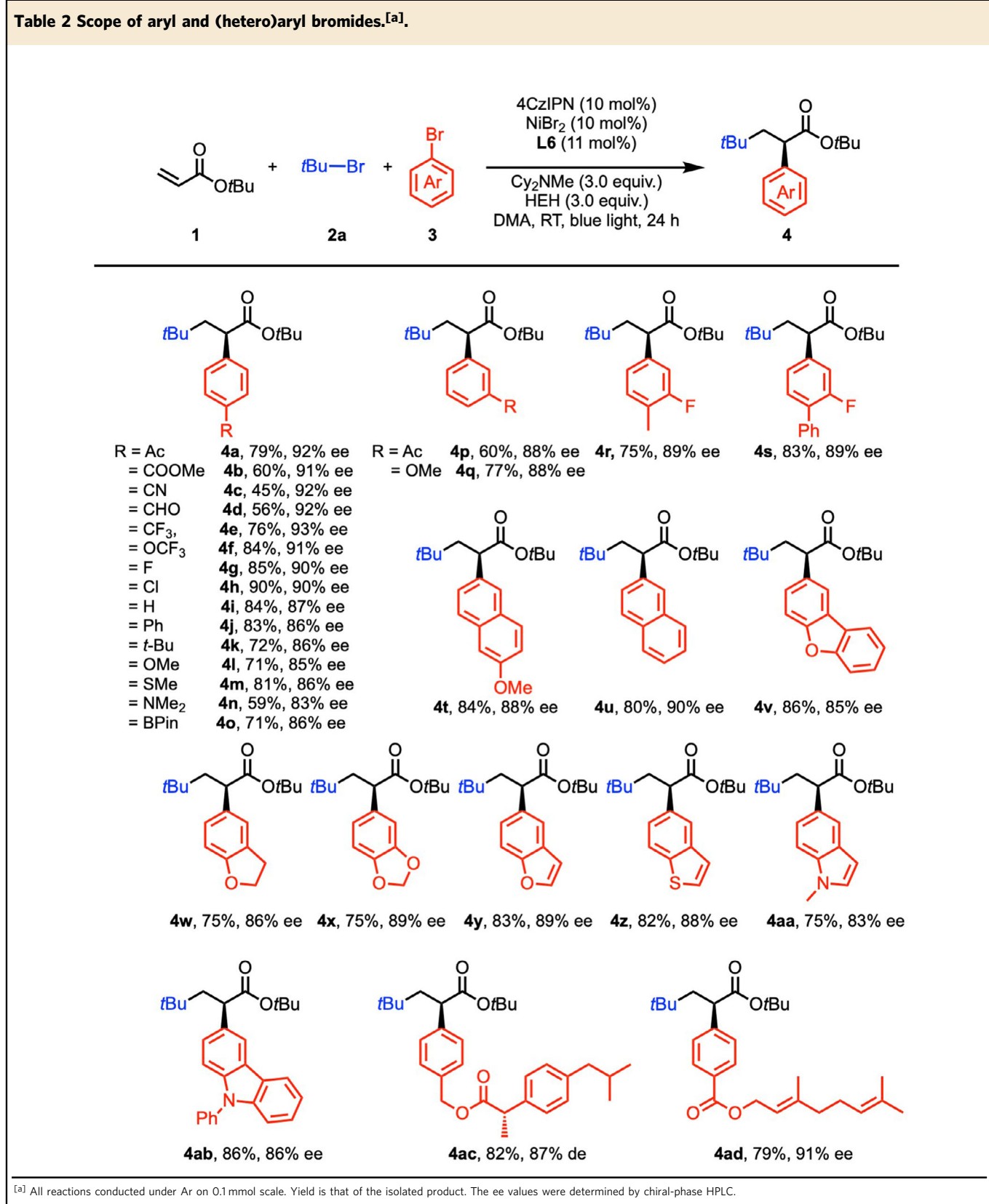

R = Ac **4a**, 79%, 92% ee
= COOMe **4b**, 60%, 91% ee
= CN **4c**, 45%, 92% ee
= CHO **4d**, 56%, 92% ee
= CF₃, **4e**, 76%, 93% ee
= OCF₃ **4f**, 84%, 91% ee
= F **4g**, 85%, 90% ee
= Cl **4h**, 90%, 90% ee
= H **4i**, 84%, 87% ee
= Ph **4j**, 83%, 86% ee
= t-Bu **4k**, 72%, 86% ee
= OMe **4l**, 71%, 85% ee
= SMe **4m**, 81%, 86% ee
= NMe₂ **4n**, 59%, 83% ee
= BPin **4o**, 71%, 86% ee

R = Ac **4p**, 60%, 88% ee
= OMe **4q**, 77%, 88% ee

**4r**, 75%, 89% ee

**4s**, 83%, 89% ee

**4t**, 84%, 88% ee

**4u**, 80%, 90% ee

**4v**, 86%, 85% ee

**4w**, 75%, 86% ee

**4x**, 75%, 89% ee

**4y**, 83%, 89% ee

**4z**, 82%, 88% ee

**4aa**, 75%, 83% ee

**4ab**, 86%, 86% ee

**4ac**, 82%, 87% de

**4ad**, 79%, 91% ee

[a] All reactions conducted under Ar on 0.1 mmol scale. Yield is that of the isolated product. The ee values were determined by chiral-phase HPLC.

(45–90%). It is noteworthy that in prior studies, aryl bromides with electron-donating groups were poor substrates in reductive cross-coupling reactions[43]. Additionally, the 4-bromo phenyl BPin, which can be used for further transformations, was also well tolerated (71% yield, 86% ee).

Aryl bromides bearing functional groups in the 3-position (Ac and OMe) were fine substrates, furnishing the products **4p** (60% yield, 88% ee) and **4q** (77% yield, 88% ee), respectively. Disubstituted aryl bromides were good cross-coupling partners, affording products **4r** with 89% ee and 75% yield. It is noteworthy

that Flurbiprofen analog **4s** and Naproxen analog **4t** were obtained in good yields (83–84%) with excellent enantioselectivities (88–89%). Unfortunately, sterically hindered 2-substituted aryl bromides were not tolerated in this three-component asymmetric reductive cross-coupling (see Supporting Information for details).

Aryl bromides with extended π-systems, such as 2-bromonaphthylene, provided product **4u** (80% yield, 90% ee). Heterocycles are important structural motifs in medicinal chemistry. Several aryl bromides containing heterocycles were, therefore, examined. 2-Bromodibenzo[b,d]furan, 5-bromo-2,3-dihydrobenzofuran, 5-bromobenzo[d][1,3]dioxole, 5-bromobenzofuran, 5-bromobenzo[b]thiophene, 5-bromo-1-methyl-1H-indole and 3-bromo-9-phenyl-9H-carbazole all exhibited good to excellent enantioselectivities (83–89%) with good yields (75–86%). To put the utility of this method to the test, ibuprofen and geraniol derivatives were subjected to the 3-component reaction. The desired products (**4ac, 4ad**) were generated in 79–82% yields and 87% de, 91% ee.

**Table 3 Scope of 3°-alkyl bromides.[a].**

**5a**, 75%, 92% ee

**5b**, 70%, 85% ee

**5c**, 71%, 85% ee

**5d**, 79%, 83% ee

**5e**, 72%, 83% ee

**5f**, 60%, 85% ee

**5g**, 72%, 83% ee

**5h**, 80%, 83% ee

**5i**, 80%, 90% ee

**5j**, 84%, 72% ee

**5k**, 83%, 82% ee

**5l**, 42%, 88% ee

[a] All reactions conducted under Ar on 0.1 mmol scale. Yield is that of the isolated product. The ee values were determined by chiral-phase HPLC.

To further explore the applications of this three-component asymmetric reductive cross-coupling method, we next examined the scope of the alkyl bromides (Table 3). Initially, acyclic tertiary alkyl bromides were examined. We were pleased to find that tertiary alkyl centers containing primary alkyl bromides, aryl, esters and ketone functional groups were compatible with the reaction conditions and afforded their corresponding products (**5a–5k**) in good to excellent yields (60–84%) and good to excellent enantioselectivities (72–92%). Another sterically hindered 3° alkyl bromide, 1-bromo-1-methylcyclohexane, was also well-tolerated, furnishing the product **l** (42% yield, 88% ee). Use of 1°- or 2°-alkylbromides gave rise to two-component aryl-alkyl cross coupling under these conditions[44], because reactions of radicals derived from these species at the nickel center are competitive with radical addition to the unsaturated ester.

To push the system beyond 3°-alkylhalides, additional optimization (see Table S2 in the Supporting Information for details) was conducted. Gratifyingly, moderate changes (aryl and alkyl iodides instead of their corresponding bromides, NiCl$_2$·glyme replacing NiBr$_2$ and a mix solvent acetone/DMA = 2:1 replacing DMA) under otherwise standard conditions enabled coupling of less substituted alkyl radicals (Table 4). Not only 1°-alkyl halides (1-iodobutane) but also 2°-alkyl halides (2-iodopropane, iodocyclopentane, iodocyclohexane, iodocycloheptane, iodocyclooctane) were tolerated, giving their corresponding products (**6a–6f**) in 15–75% yield and 76–92% ee. Other electron-deficient alkenes, such as *N*-phenylacrylamide, was tested under the standard conditions and afforded the product **6g** (40% yield, 33% ee). It is noteworthy that the electron-rich vinyl benzoate was also tolerated, albeit with diminished yield and ee (**6h**, 45% yield, 50% ee).

**Mechanistic Studies.** We desired to gain insight into the reaction pathway. Based on our previous work[19], a detail analysis of the model reaction was conducted (Fig. 2a). The product **4a** was isolated in 79% yield with 92% ee. The fate of the HEH was the

**Table 4 Scope of 1° and 2°-alkyl iodides and alkenes.[a]**

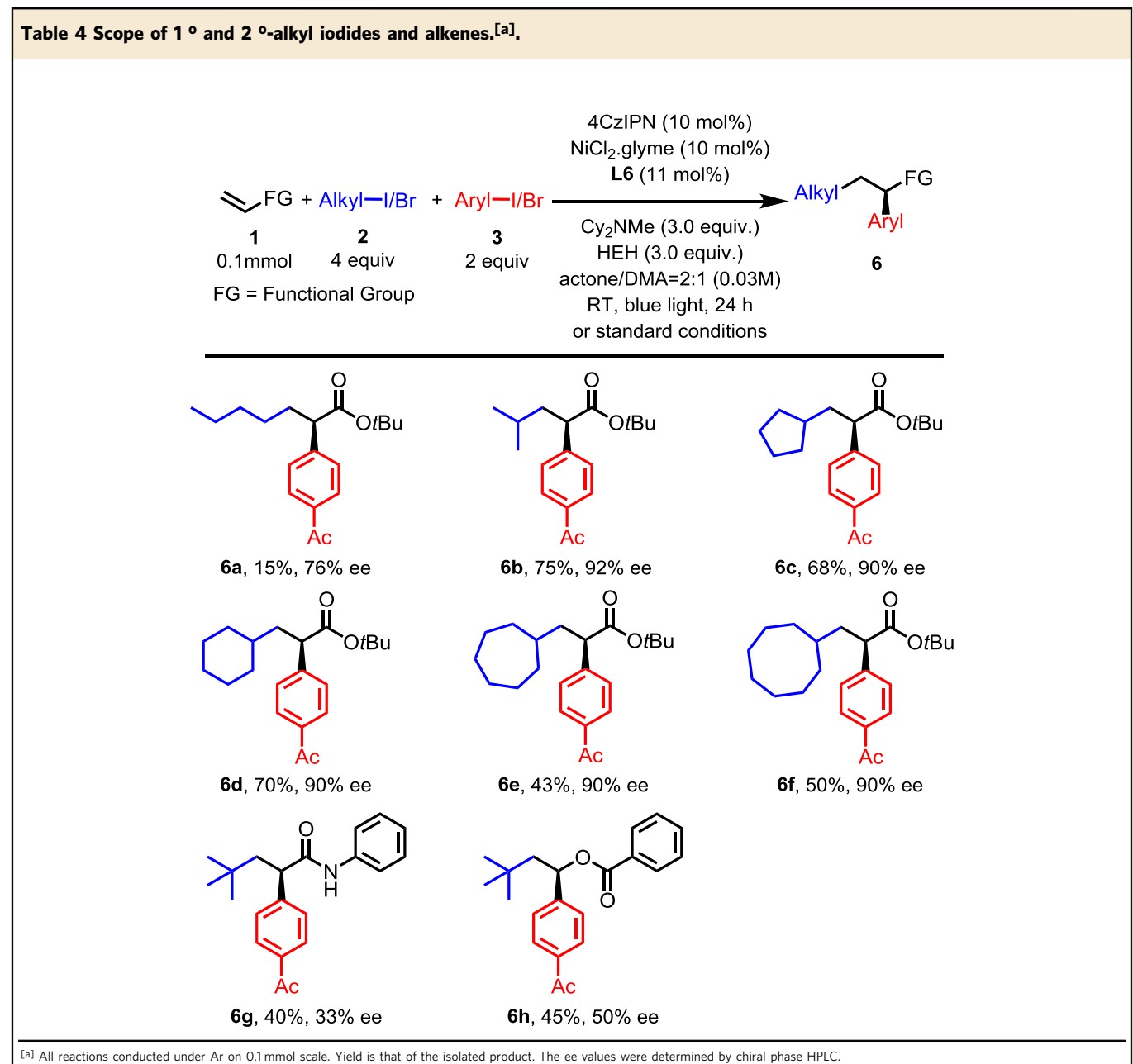

[a] All reactions conducted under Ar on 0.1 mmol scale. Yield is that of the isolated product. The ee values were determined by chiral-phase HPLC.

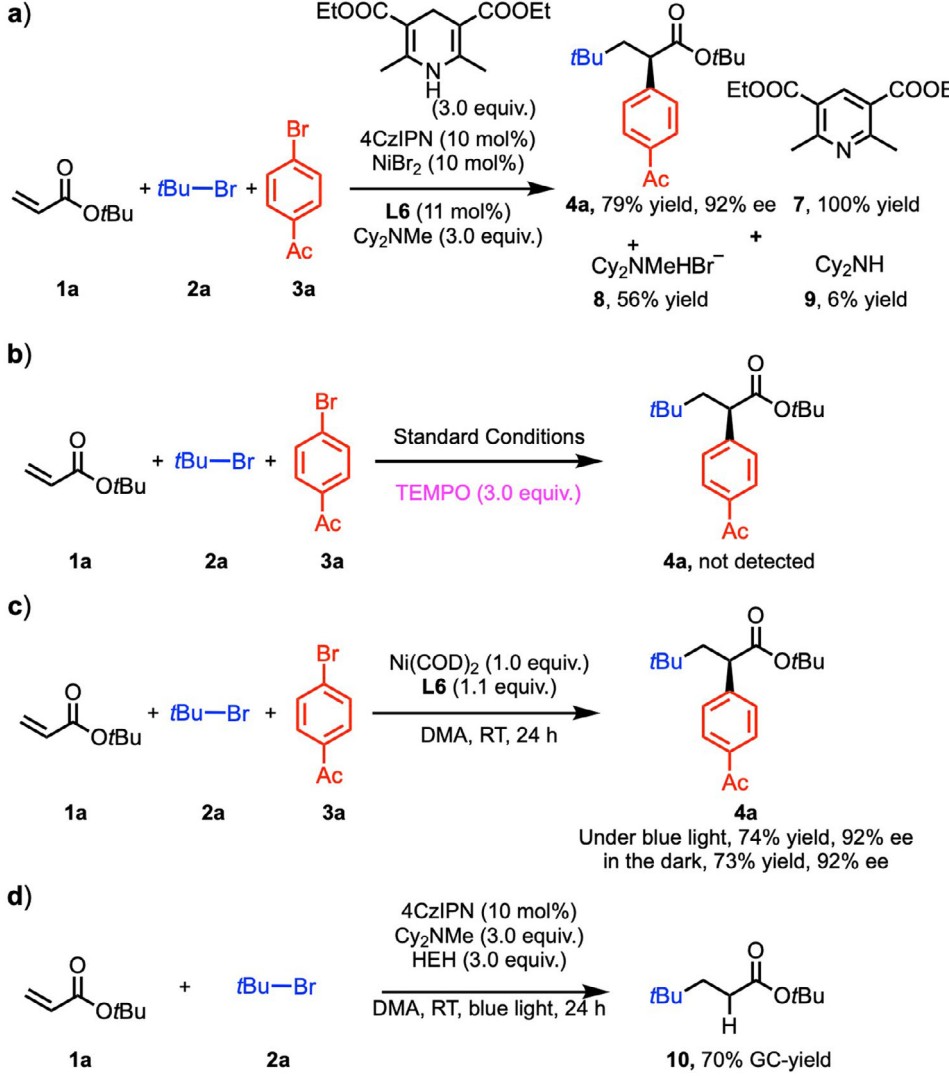

**Fig. 2 Mechanistic experiments. a** Detail analysis of the model reaction. **b** Radical trapping reaction via TEMPO. **c** Stoichiometric studies. **d** Intermediate verification reaction.

expected pyridine (isolated in 100% yield relative to HEH), derived from donation of 2 electrons and two protons. In the presence of HEH, Cy$_2$NMe most likely acted as a base, as determined by the isolation of Cy$_2$NMe·HBr (**8**, 56% yield). Only about 6% of the demethylated product, Cy$_2$NH, was detected after workup. The demethylated amine (**9**) likely arises from oxidation of the amine by *4CzIPN to the amine radical cation, loss of H• to generate the iminium ion, and hydrolysis by advantageous water during the reaction or upon workup[40].

Additional experiments were then carried out to probe the reaction pathway. To explore the possibility of radical inter-mediates, the reaction was conducted under the standard conditions with the addition of the radical trap TEMPO. The presence of TEMPO shut down the formation of **4a**, consistent with the involvement of radicals (Fig. 2b).

Stoichiometric studies, wherein NiBr$_2$ was replaced by 1.0 equiv Ni(COD)$_2$ in the presence of 1.1 equiv **L6** but in the absence of 4CzIPN, HEH and Cy$_2$NMe were carried out. This experiment was conducted both with and without blue light irradiation. With the light on **1** reacted with **2a** and **3a** to form cross-coupled product **4a** (74% yield, 92% ee) (Fig. 2c). Likewise, when the reaction was conducted without irradiation, the cross-coupled product **4a** was observed (73% yield, 92% ee). It is noteworthy that the *ee* of these

stoichiometric reactions are identical to that observed under the standard catalytic conditions (Table 1, entry 1). These observations suggest that 1) the enantiodetermining step in the catalytic and stoichiometric reactions (Fig. 2b) are identical, and do not involve the photoredox cycle and 2) *tert*-butyl radical could be formed via SET reduction from the nickel catalyst (either Ni$^0$ or Ni$^I$).

To probe the function of the photoredox cycle in this system, we performed the model reaction without Ni/**L6** and bromo-benzene (Fig. 2d). The radical addition/HAT quenching product (**10**) was obtained in 70% AY. This result indicates that in addition to reducing the Ni catalyst, the photoredox catalyst can undergo SET to the *tert*-butyl bromide to generate the *tert*-butyl radical. The radical then undergoes the Giese-type addition to the acrylate followed by abstraction of H• from HEH. Given that the olefin difunctionalization also proceeds via an α-carbonyl radical, it must be that the α-carbonyl radicals add to L*Ni(Ar)Br faster than they undergo HAT from HEH.

On the basis of related studies[45], and the mechanistic experiments above, we present a proposed dual catalytic pathway for the enantioselective reductive three-component alkyl arylation reaction of alkenes with tertiary alkyl- and aryl-bromides (Fig. 3). The active (BiIM)Ni$^0$ [$E1/2^{red}$ (Ni$^{II}$/Ni$^0$) = −1.2 V vs. SCE] is generated in situ in two SET steps by the reduced photocatalyst

**Fig. 3 Plausible catalytic reaction pathway.** Key steps include oxidative addition of Ar–Br to Ni(0), radical generation by SET from nickel to the alkyl bromide, Giese addition of the alkyl radical to the acrylate, trapping of the α-carbonyl radical by Ni(II) to give Ni(III), which undergoes reductive elimination to give the product. The photocatalytic cycle involves reduction of 4CzIPN to 4CzIPN⁻, which is proposed to reduce the Ni catalyst to Ni(0).

4CzIPN$^{\bullet-}$ ($E1/2^{red} = -1.21$ V vs. SCE)[46–49]. The resulting (BiIM)Ni$^0$ catalyst oxidatively adds the aryl bromide to give the (BiIM)Ni(Ar)Br complex[50]. The alkyl bromide undergoes SET by either the Ni or reduced 4CzIPN$^{\bullet-}$ (as described above[45]) to generate the alkyl halide radical anion, which looses bromide to generate the tertiary radical. Addition of the tertiary radical to the acrylate forms the α-carbonyl radical that is trapped by the (BiIM)Ni(Ar)Br complex to form the reactive Ni$^{III}$ species. The resulting Ni$^{III}$ intermediate[51] then undergoes rapid reductive elimination to give the product **4a**. In the photoredox cycle, photoexcited *4CzIPN ($\tau = 5.1 \pm 0.5$ μs)[52] reacts with HEH to afford 4CzIPN$^{\bullet-}$. The resulting 4CzIPN$^{\bullet-}$ is oxidized by the nickel catalyst (Ni$^{II}$/Ni$^{I}$) and/or **2a** to regenerate 4CzIPN.

In conclusion, we have developed an example of dual nickel/organic photoredox catalyzed asymmetric reductive difunctionalization of acrylates with alkyl- and aryl-bromides to generate NSAID derivatives. The advantages of this reductive protocol are that it uses two bench stable and commercially available electrophiles and it is broadly tolerant of functional groups, avoids preformed moisture- and/or air-sensitive organometallic reagents, and uses an organic reductant rather than the metal reductants used in most cross-electrophile coupling reactions.

## Methods

**General procedure 1**. To an oven-dried vial equipped with a stir bar was added NiBr$_2$ (2.2 mg, 0.01 mmol), **L6** (5.4 mg, 0.011 mmol) and DMA (3.0 mL) under an argon atmosphere inside a glove box at RT to give a light green solution. After 30 min at RT, olefin (0.1 mmol, 1.0 equiv.), 3°-alkyl bromide (0.4 mmol, 4.0 equiv.) and aryl bromide (0.2 mmol, 2.0 equiv.) were added. Next, Cy$_2$NMe (58.5 mg, 64 μL, 0.3 mmol, 3.0 equiv.), HEH (75.9 mg, 0.3 mmol, 3.0 equiv.) and 4CzIPN (8.0 mg, 0.01 mmol) were added. Once HEH and 4CzIPN were added, the solution turned from light green to yellow. The vial was sealed with a cap and removed from the glove box. The reaction mixture was stirred at RT under blue light. After 24 h, the color of the reaction mixture changed back to light green. The vial was opened to air and EtOAc (10 mL) was added. The resulting solution was then washed with brine (5 mL × 5) and the organic layer separated and dried over anhydrous Na$_2$SO$_4$, filtered and concentrated to give the crude product. The crude residue was purified by flash column chromatography to afford the corresponding product.

**General procedure 2**. To an oven-dried vial equipped with a stir bar was added NiCl$_2$•glyme (2.2 mg, 0.01 mmol), **L6** (5.4 mg, 0.011 mmol) and acetone/DMA (2.0/1.0, v/v, 3.0 mL) under an argon atmosphere inside a glove box at RT, giving a light green solution. After 30 min at RT, tert-butyl acrylate (0.1 mmol, 1.0 equiv.), 1° or 2°-alkyl iodides (0.4 mmol, 4.0 equiv.), 1-(4-iodophenyl)ethan-1-one (0.2 mmol, 2.0 equiv.), Cy$_2$NMe (58.5 mg, 64 μL, 0.3 mmol, 3.0 equiv.), HEH (75.9 mg, 0.3 mmol, 3.0 equiv.) and 4CzIPN (8.0 mg, 0.01 mmol) were added. Once HEH and 4CzIPN were added, the

solution turned from light green to yellow. The vial was sealed with a cap and removed from the glove box. The reaction mixture was stirred at RT under blue light. After 24 h, the color of the solution changed back to light green. The vial was opened to air and EtOAc (10 mL) was added to the solution, which was then washed with brine (5 mL × 5). The organic layer was dried over anhydrous Na$_2$SO$_4$, filtered and concentrated to give the crude product. The crude residue was purified by flash column chromatography to afford the corresponding product.

## Data availability

The detailed experimental procedures and characterization of compounds data generated in this study are provided in Supplementary Information. The authors declare that all other data supporting the findings of this study are available within the article and Supplementary Information files, and also are available from the corresponding author upon request.

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

## Acknowledgements
We acknowledge the National Natural Science Foundation of China (22071107 and 21801128 to J.M.), Natural Science Foundation of Jiangsu Province, China (BK20170965 to J.M.), and Nanjing Tech University (BK20211588 and 39837112) for financial support. P.J.W. thanks the US National Science Foundation (CHE-1902509).

## Author contributions
P.Q. and H.G. are contributed equally to this work and mechanistic study with the help of Y.W., Q.L., F.Z. and D.X. The project was conceived by J.M. and P.Q. with help from P.J.W. The project was directed by J.M. and the manuscript was written by H.G., J.M. and P.J.W.

## Competing interests
The authors declare no competing interests.

## Additional information



