## [Peer Review File · Nature Communications]

Catalytic Enantioselective Reductive Domino Alkyl Arylation of Acrylates via Nickel/Photoredox CatalysisREVIEWER COMMENTS

Reviewer #1 (Remarks to the Author):

This manuscript authored by Mao, Walsh and co-workers describes a nickel/photoredox cocatalyzed enantioselective reductive alkyl arylation of tert-butyl acrylate with tertiary alkyl bromides and aryl bromides. Enantioenriched α -aryl propionic acid derivatives are delivered, which are an important class of nonsteroidal anti-inflammatory drugs. This three-component reductive coupling avoids preformed organometallic reagents and organic base is used as the reductant avoiding the use of stoichiometric metal reductants. Mechanistically, this difunctionalization of acrylates proceeds via initial SET from the nickel catalyst to an alkyl bromide to generate a tertiary radical. Giese-type addition of this alkyl radical to an acrylate to generate an α -carbonyl radical. This α -carbonyl radical adds to (BiIM)Ni(II)Ar(Br) generated through oxidative addition of Ni(0) with an aryl bromide. The resultant Ni(III) intermediate undergoes reductive elimination to afford the enantioenriched final product.

This mechanistically novel strategy expands the scope of enantioselective C(sp²)-C(sp³) reductive couplings, which makes a definite contribution to the synthetic chemist's toolbox. This work is also a significant contribution to enantioselective metallaphotoredox catalysis, as well as nickel catalysis. The scope and limitations of the reaction were studied and showed that it was amenable to a range of substrates. The results reported in this paper are both significant and intriguing. This reviewer would like to recommend acceptance of this work after the following issues are addressed.

- 1) Two organic bases (Hantzsch ester and Cy2NMe, 3 equiv each) are employed in this reaction. The authors conclude that Hantzsch ester was a terminal reductant for the reaction and Cy2NMe primarily acted as a base. Any evidence for this statement? Can other amines replace Cy2NMe as a base?
- 2) For substrate scope of acrylates, only tert-butyl acrylate was tested. How about other electron-deficient alkenes?
- 3) The authors discuss the work of Doyle and Weix (Scheme 1, c). However, no information is found in Scheme 1c.
- 4) Page 5, "In this case, it is possible that the Cy2NMe plays the role of reducing agent, reducing agent, albeit with much reduced efficiency." Two "reducing agent".
- 5) Please define TDAE when it first appears in Page 5.
- 6) For substrate 4ac, two chiral centers present in this compound. The "ee" is not suitable for describing optical purity of this substrate. The "de" is better.
- 7) Scheme 4b, please delete "H" from "tBu-BrH".
- 8) Ref 3 of main text and ref 8 in SI are incomplete. Article number for ref 22 of main text is wrong. A lot of journal names are not shown in abbreviation. Please double check and modified all the references both in main text and SI.

Reviewer #2 (Remarks to the Author):

In this manuscript, Walsh and Mao describe nickel- and photoredox-catalyzed asymmetric three-component alkyl-arylation of acrylates with alkyl bromides and aryl bromides in the presence of Hantzsch ester to give enantioenriched α -aryl esters with moderate to high ees. The authors of this paper carried out the three-component transformation with a lot of aryl and tertiary alkyl bromides and some mechanistic studies to elucidate the reaction mechanism. Conceptually, this reaction takes the same dual catalytic reductive dicarbofunctionalization strategy reported by Martin (Angew Chem 2020, 59, 4370), achieving the challenging asymmetric alkylarylation of alkenes via dual catalysis. As described in the introduction of this manuscript, the content of the present manuscript is complementary to the work disclosed by Gutierrez and Chu (J Am Chem Soc 2020, 142, 9604). Regrettably, only moderate to high ees were obtained in almost all cases in the present reaction (most of them are below 90% ee), a fatal weak point for a catalytic asymmetric reaction. Additionally, the substrate scope of this present reaction is restricted to electron-deficient acrylates. The present reaction also suffers from a bewildering mechanism. Two organic bases were used and HEH was taken for granted as the stoichiometric reductant, which seemed untenable based on the fact that 32% of 4a was still observed in the absence of HEH (entry 11).

These SET processes for the elucidation of the mechanism shown in Figure 1 also seem ungrounded and confusing. As a result, this reviewer believes that the content of the present manuscript provides some new results, however, the novelty and the synthetic applicability of this paper are not suitable for publication in Nat Commun the present form.

Below please find our reply to the reviewer's comments.

COMMENTS

Reviewer 1: This manuscript authored by Mao, Walsh and co-workers describes a nickel/photoredox cocatalyzed enantioselective reductive alkyl arylation of tert-butyl acrylate with tertiary alkyl bromides and aryl bromides. Enantioenriched α -aryl propionic acid derivatives are delivered, which are an important class of nonsteroidal anti-inflammatory drugs. This three-component reductive coupling avoids preformed organometallic reagents and organic base is used as the reductant avoiding the use of stoichiometric metal reductants. Mechanistically, this difunctionalization of acrylates proceeds via initial SET from the nickel catalyst to an alkyl bromide to generate a tertiary radical. Giese-type addition of this alkyl radical to an acrylate to generate an α -carbonyl radical. This α -carbonyl radical adds to (BiIM)Ni(II)Ar(Br) generated through oxidative addition of Ni(0) with an aryl bromide. The resultant Ni(III) intermediate undergoes reductive elimination to afford the enantioenriched final product. This mechanistically novel strategy expands the scope of enantioselective C(sp²)-C(sp³) reductive couplings, which makes a definite contribution to the synthetic chemist's toolbox. This work is also a significant contribution to enantioselective metallaphotoredox catalysis, as well as nickel catalysis. The scope and limitations of the reaction were studied and showed that it was amenable to a range of substrates. The results reported in this paper are both significant and intriguing. **This reviewer would like to recommend acceptance of this work after the following issues are addressed.**

Comments: 1) Two organic bases (Hantzsch ester and Cy₂NMe, 3 equiv each) are employed in this reaction. The authors conclude that Hantzsch ester was a terminal reductant for the reaction and Cy₂NMe primarily acted as a base. Any evidence for this statement?

Response: In our previous work (*Angew. Chem. Int. Ed.* 2020, 59, 5172-5177), similar conditions [two organic bases (Hantzsch ester and Cy₂NMe, 3 equiv each)] were employed. The fate of the HEH was the expected pyridine (isolated in 100% yield relative to HEH), derived from donation of 2 electrons and two protons. The Cy₂NMe largely acted as a base in the reaction, as determined by the isolation of 55% of the total Cy₂NMe as Cy₂NMe·HX (X = Cl or I). Building on these results, we concluded that Hantzsch ester was a terminal reductant for the reaction and Cy₂NMe primarily acted as a base.

The Reviewer continued: Can other amines replace Cy₂NMe as a base?

Response: To verify this question, we tried several different amines and two inorganic bases (See

below). From the results, we can conclude that other amines such as Et₃N (entry 2, 58% yield, 91% ee), TMEDA (entry 3, 16% yield, 72% ee) and *N,N*-dimethylethylenediamine (entry 6, 50% yield, 91% ee) were also feasible for this reaction, but the yield and ee were diminished. It should be noted that inorganic base Na₂CO₃ could replace Cy₂NMe to complete this reaction, which also suggests that Hantzsch ester was a terminal reductant and Cy₂NMe primarily acted as a base. These results were added to the Supporting Information (Page S62).

Entry	Base (3 equiv.)	GC yield (%)	ee (%)
1	Cy ₂ NMe	82	92
2	Et ₃ N	58	91
3	TMEDA	16	72
4	N,N -Dimethyl-1,2-ethanediamine	Trace	--
5	Triethanolamine	Trace	--
6	N,N -Dimethylethylenediamine	50	91
7	Na ₂ CO ₃	60	91
8	Cs ₂ CO ₃	Trace	--

Comments: 2) For substrate scope of acrylates, only tert-butyl acrylate was tested. How about other electron-deficient alkenes?

Response: We appreciate the reminder from the reviewer. First of all, several different acrylates were tested (see below). Methyl acrylate (entry 1), ethyl acrylate (entry 2), benzyl acrylate (entry 4), phenyl acrylate (entry 5) and cyclohexyl acrylate (entry 6) were tolerable in this reaction, but the yield and ee were diminished. These results were added to the revised Supporting Information (Page S61).

Entry	R	Isolated yield (%)	ee (%)
1	Me	25	74
2	Et	28	65
3	tBu	82	92
4	Bn	31	55
5	Ph	35	47
6	Cy	49	59

Response continued: Additionally, other electron-deficient alkenes such as *N*-phenylacrylamide (40% yield, 33% ee), *tert*-butyl (*E*)-but-2-enoate, *tert*-butyl methacrylate, furan-2(5*H*)-one, (methylsulfonyl)ethane and (vinylsulfonyl)benzene were tested and the results are show below. It should be noted that the electron-rich alkene vinyl benzoate (45% yield, 50% ee) was also tolerated in this reaction, but the yield and ee diminished. Some of these results were added to the revised manuscript while the rest were put in the revised Supporting Information (Page S66).

Comments: 3) The authors discuss the work of Doyle and Weix (Scheme 1, c). However, no information is found in Scheme 1c.

Response: Thanks for the reviewer's suggestion. A lot of fantastic works were done by Doyle and Weix each in this field. However, the theme of Scheme 1 was "*Transition-Metal-Catalyzed Asymmetric Coupling Reactions to Synthesize NSAID Derivatives*" and they do not have similar examples.

Comments: 4) Page 5, "In this case, it is possible that the Cy₂NMe plays the role of reducing agent, reducing agent, albeit with much reduced efficiency." Two "reducing agent".

Response: This typo has been corrected.

Comments: 5) Please define TDAE when it first appears in Page 5.

Response: We have added it in the revised manuscript.

Comments: 6) For substrate 4ac, two chiral centers present in this compound. The “ee” is not suitable for describing optical purity of this substrate. The “de” is better.

Response: Thanks for the reviewer’s suggestion, we have revised it in the revised manuscript.

Comments: 7) Scheme 4b, please delete “H” from “tBu-BrH”.

Response: This typo has been corrected.

Comments: 8) Ref 3 of main text and ref 8 in SI are incomplete. Article number for ref 22 of main text is wrong. A lot of journal names are not shown in abbreviation. Please double check and modified all the references both in main test and SI.

Response: We have revised it in the revised manuscript.

Reviewer 2: In this manuscript, Walsh and Mao describe nickel- and photoredox-catalyzed asymmetric three-component alkyl-arylation of acrylates with alkyl bromides and aryl bromides in the presence of Hantzsch ester to give enantioenriched α -aryl esters with moderate to high ees. The authors of this paper carried out the three-component transformation with a lot of aryl and tertiary alkyl bromides and some mechanistic studies to elucidate the reaction mechanism. Conceptually, this reaction takes the same dual catalytic reductive dicarbofunctionalization strategy reported by Martin (*Angew Chem* 2020, 59, 4370), **achieving the challenging asymmetric alkylarylation of alkenes via dual catalysis.**

Response: Thanks very much for the reviewer’s recognition. The inspiration of this work is mainly from our previous work (*Angew. Chem. Int. Ed.* 2020, 59, 5172-5177), the first enantioselective dual nickel and organic photoredox catalyzed reductive cross coupling between α -chloro esters and aryl iodides to afford α -aryl esters.

The reviewer continues: As described in the introduction of this manuscript, the content of the present manuscript is complementary to the work disclosed by Gutierrez and Chu (*J Am Chem Soc* 2020, 142, 9604). Regrettably, only moderate to high ees were obtained in almost all cases in the present reaction (most of them are below 90% ee), a fatal weak point for a catalytic asymmetric reaction. Additionally, the substrate scope of this present reaction is restricted to electron-deficient

acrylates.

Response: We appreciate the reviewer's detailed analysis of the context of this chemistry. While it is true that the ee's could be higher, this work still represents the state of the art.

The reviewer continues: The present reaction also suffers from a bewildering mechanism. Two organic bases were used and HEH was taken for granted as the stoichiometric reductant, which seemed untenable based on the fact that 32% of 4a was still observed in the absence of HEH (entry 11). These SET processes for the elucidation of the mechanism shown in Figure 1 also seem ungrounded and confusing.

Response: As noted above, we have verified the mechanism in our previous work (*Angew. Chem. Int. Ed.* 2020, 59, 5172-5177).

The reviewer continues: As a result, this reviewer believes that the content of the present manuscript provides some new results, however, the novelty and the synthetic applicability of this paper are not suitable for publication in Nat Commun the present form.

Response: Our submitted manuscript was restricted to 3°-alkylbromides. Generally speaking, use of 1°- or 2°-alkylbromides gave rise to two-component aryl-alkyl cross coupling (*Org. Lett.* 2016, 18, 4012-4015), because reaction of radicals derived from these species react at the nickel center and are competitive with radical addition to the unsaturated ester.

In the revised manuscript, after an additional optimization (see Table S2 in the Supporting Information for details), we were please to find that moderate changes (aryl and alkyl iodides instead of their corresponding bromides, NiCl₂·glyme replacing Ni(COD)₂ and a mix solvent acetone/DMA=2:1 replacing DMA) in standard conditions could realize the scope of 1° and 2°-alkylhalides (see below).

Additionally, other electron-deficient alkenes such as diethyl vinylphosphonate and *N*-phenylacrylamide were tested under the standard conditions and afforded the product **6g** (51% yield, 93% ee) and **6h** (40% yield, 33% ee), respectively. It is noteworthy that electron-rich alkene vinyl benzoate was also tolerated in this reaction, although it remains challenging (product **6i**, 45% yield, 50% ee).

REVIEWER COMMENTS

Reviewer #1 (Remarks to the Author):

The revised manuscript has been reviewed and the issues raised by this reviewer have fully addressed. This revised manuscript can be accepted at this stage.

Reviewer #2 (Remarks to the Author):

Comments:

In this manuscript, Mao, Walsh, and co-workers describe a nickel/photoredox dual catalyzed enantioselective reductive alkylation of acrylates with alkyl bromides and aryl bromides, providing facile access to a series of enantioenriched α -aryl propionic acids. This is a revised article. It's great to see that the authors have expanded the alkyl scope to secondary and even primary alkyl bromides, although the scope of acrylates is still considerably restricted. Considering the current synthetic challenges regarding this topic, I would like to recommend its publication in Nature Communications only if a revised mechanism could be provided.

I am not convinced by the authors' claim on the role of HE and Cy2NMe. It could be corrected that in the presence of HE, Cy2NMe acts most likely as a base and can be replaced by other organic or inorganic bases. While control reactions clearly indicated that in the absence of HE, a 32% yield of product was observed. In this case, only Cy2NMe can act as the reductant. That been said, the authors might consider the possibility of SET oxidation of Cy2NMe by the photoexcited catalyst in their proposed mechanism (Scheme 6).

In Supplementary Information, all subtitles for HPLC spectra are unreadable. Please check carefully.

Reviewer 1: The revised manuscript has been reviewed and the issues raised by this reviewer have fully addressed. This revised manuscript can be accepted at this stage.

Response: Thanks very much for the reviewer's recognition.

Reviewer 2: In this manuscript, Mao, Walsh, and co-workers describe a nickel/photoredox dual catalyzed enantioselective reductive alkylarylation of acrylates with alkyl bromides and aryl bromides, providing facile access to a series of enantioenriched α -aryl propionic acids. This is a revised article. It's great to see that the authors have expanded the alkyl scope to secondary and even primary alkyl bromides, although the scope of acrylates is still considerably restricted. Considering the current synthetic challenges regarding this topic, **I would like to recommend its publication in Nature Communications only if a revised mechanism could be provided.**

Comments 1): I am not convinced by the authors' claim on the role of HE and Cy₂NMe. It could be corrected that in the presence of HE, Cy₂NMe acts most likely as a base and can be replaced by other organic or inorganic bases.

Response: Thanks for the suggestion. We conducted detail analysis of the model reaction (Fig. 2A in the revised manuscript). The product **4a** was isolated in 79% yield with 92% ee. The fate of the HEH was the expected pyridine (isolated in 100% yield relative to HEH), derived from donation of 2 electrons and two protons. **In the presence of HEH, Cy₂NMe acted most likely as a base,** as determined by the isolation of Cy₂NMe•HBr (**8**, 56% yield). Only about 6% of the demethylated product, Cy₂NH, was detected after workup. The demethylated amine likely arises from oxidation of the amine by *4CzIPN to the amine radical cation, loss of H⁺ to generate the iminium ion, and hydrolysis by advantageous water during the reaction or upon workup (Nat. Commun. 9, 4936 (2018)). We have revised the description of the role of HE and Cy₂NMe in the revised manuscript (Mechanistic Studies). As instructed by the reviewer, we used “in the presence of HE, Cy₂NMe most likely acted as a base” replacing “the Cy₂NMe primarily acted as base” and highlight it.

The Reviewer continued: While control reactions clearly indicated that in the absence of HE, a 32% yield of product was observed. In this case, only Cy₂NMe can act as the reductant. That been said, the authors might consider the possibility of SET oxidation of Cy₂NMe by the photoexcited catalyst in their proposed mechanism (Scheme 6).

Response: We have considered the possibility of SET oxidation of Cy₂NMe and added it in the revised manuscript (Scheme 6, following the editor's instructions, we re-numbered the scheme to Fig 3).

Comments 2): In Supplementary Information, all subtitles for HPLC spectra are unreadable. Please check carefully.

Response: We have revised all the subtitles in the revised Supplementary Information.

REVIEWERS' COMMENTS

Reviewer #2 (Remarks to the Author):

All concerns have been addressed. And I would like to recommend its publication in Nature Communications.

Reviewer 2: All concerns have been addressed. And I would like to recommend its publication in Nature Communications.

Response: Thanks very much for the reviewer's recognition.

We thank the reviewers for their helpful suggestion. With the changes we have made, we hope that the manuscript is now acceptable for publication in *Nature Communications*.